# Ligand-Enhanced Zero-Valent Iron for Organic Contaminants Degradation: A Mini Review

**Qi Chen [1,2], Minghua Zhou [1,2,*], Yuwei Pan [3] and Ying Zhang [1,2,*]**

[1] Key Laboratory of Pollution Process and Environmental Criteria, Ministry of Education, College of Environmental Science and Engineering, Nankai University, Tianjin 300350, China
[2] Tianjin Key Laboratory of Environmental Technology for Complex Trans-Media Pollution, College of Environmental Science and Engineering, Nankai University, Tianjin 300350, China
[3] College of Biology and the Environment, Nanjing Forestry University, Nanjing 210037, China
[*] Correspondence: zhoumh@nankai.edu.cn (M.Z.); zhangying04@nankai.edu.cn (Y.Z.)

**Abstract:** For nearly three decades, zero-valent iron (ZVI) has been used in wastewater treatment and groundwater and soil remediation. ZVI can degrade contaminants by reactions of adsorption, redox, and co-precipitation. It can also react with oxidants like hydrogen peroxide, persulfate, and ozone to produce highly reactive radicals that can rapidly remove and even mineralize organic contaminants. However, the application of ZVI is also limited by factors such as the narrow pH range and surface passivation. The addition of chelating agents such as nitrilotriacetic acid (NTA), ethylenediaminetetraacetic acid (EDTA), or citrate to the ZVI-based processes has been identified to greatly increase the iron stability and improve the efficiency of contaminant degradation. From the perspective of commonly used organic and inorganic chelating agents in ZVI applications, the review addresses the current status of ligand-enhanced ZVI degradation of organic contaminants, illustrates the possible reaction mechanism, and provides perspectives for further research.

**Keywords:** chelating agents; zero-valent iron; Fenton-like process; oxidative species; reaction mechanism

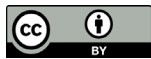

## 1. Introduction

Advanced oxidation processes (AOPs) have been widely used to degrade recalcitrant organic contaminants in water/wastewater, such as pesticides, pharmaceuticals, and personal care products (PPCPs) [1-3]. Iron is abundant in the earth's crust, accounting for up to 4.75% of the total content [4]. Due to its abundance, low price, environmental friendliness, simple remediation process, and capacity to produce non-toxic Fe(II) for sub-reactions, iron-based technology is commonly employed in the environmental remediation of soil and water [5-7]. Zero-valent iron (ZVI) technology as a promising AOP has been receiving increasing attention. ZVI with a standard redox potential $E^0$ of -0.44 V [8] can remove pollutants by reduction, oxidation, and co-precipitation (see Figure 1 [8]) [9]. It can also activate hydrogen peroxide ($H_2O_2$), oxygen ($O_2$), persulfate, ozone ($O_3$), etc., to produce reactive species such as hydroxyl radicals ($\cdot OH$), superoxide radicals ($O_2^{\cdot-}$), singlet oxygen ($^1O_2$), and sulfate radicals ($SO_4^{\cdot-}$). For example, ZVI-mediated oxygen activation occurs in the following reactions (Equations (1)–(5)).

$$Fe^0 + O_2 + 2H^+ \rightarrow Fe^{2+} + H_2O_2 \tag{1}$$

$$Fe^{2+} + O_2 \rightarrow Fe^{3+} + O_2^{\cdot-} \tag{2}$$

$$Fe^{2+} + O_2^{\cdot-} + 2H^+ \rightarrow Fe^{3+} + H_2O_2 \tag{3}$$

$$Fe^{2+} + H_2O_2 \rightarrow Fe^{3+} + OH^- + \cdot OH \tag{4}$$

$$2Fe^0 + O_2 + 4H^+ \rightarrow 2Fe^{2+} + 2H_2O \tag{5}$$

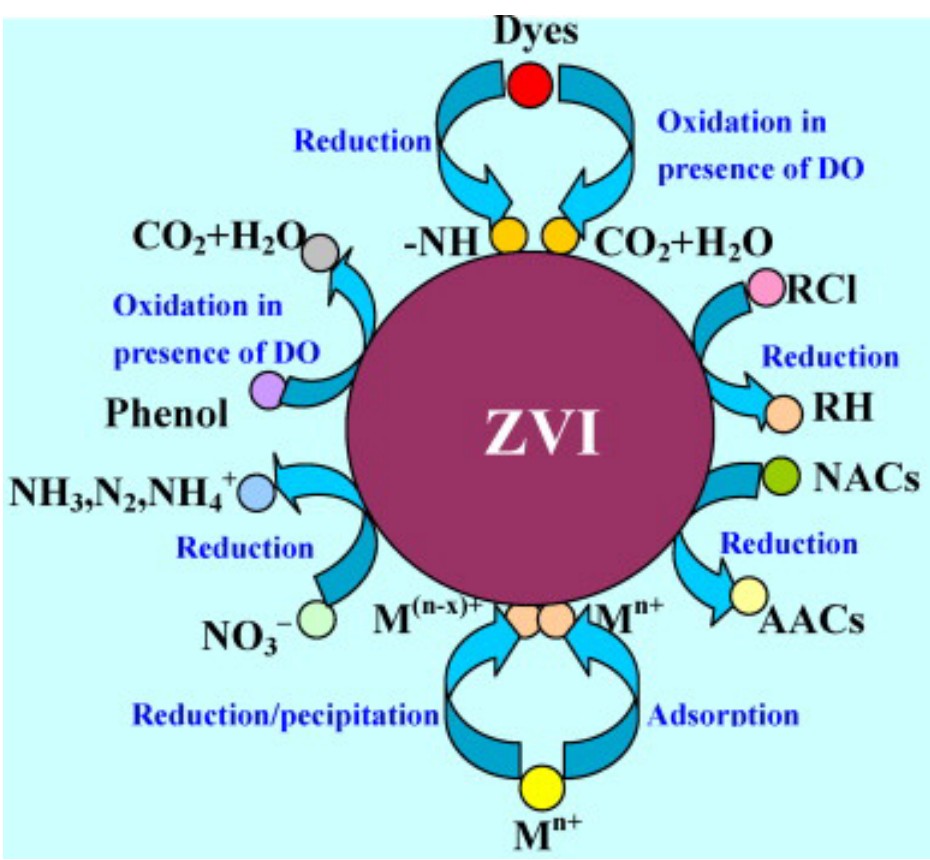

**Figure 1.** Schematic diagram for pollutant removal in ZVI systems (Reprinted from Fu et al. [8] with the permission from Elsevier).

The reactive species generated are capable of mineralizing organic macromolecules and significantly increasing the efficiency of pollutant removal. However, there are still many limitations of ZVI in the actual remediation process, such as (i) narrow pH working range, (ii) surface agglomeration, resulting in reduced site activity, and (iii) corrosion products causing passivation on the surface, which affects the electron transfer efficiency and other unfavorable factors [10-12]. Consequently, studies on the modification of ZVI have particularly increased in recent years. Guan et al. [12] summarized the countermeasures proposed between 1994 and 2014 to overcome or improve the defects of ZVI technology, which mainly consisted of pretreatment operations like acid washing of ZVI, synthesis of nanoscale ZVI or bimetals, and the addition of organic ligands or ions. Chelating agents are compounds that have several coordination sites and bond with metal ions [13]. When the chelating agent is added to the ZVI system, it first adsorbs to the surface of ZVI with the formation of an inner sphere iron complex (Fe-L) [14]. Among these, ethylenediaminetetraacetic acid (EDTA), nitrilotriacetic acid (NTA), and tetrapolyphosphate (TPP)

have been recently introduced into the ZVI systems as a hot topic for environmental pollution control. Sedlak et al. [15] reported that organic ligands like EDTA, NTA, and oxalic acid boosted the output of formaldehyde in methanol oxidation. Zhang's group [16] investigated the effect of TPP on the nZVI/$O_2$ system and the modified process achieved complete degradation of atrazine.

The mechanism of ligand-enhanced processes can be divided into two parts: heterogeneous and homogeneous processes. In the heterogeneous process, the formation of surface-bound complexes can facilitate the separation of iron ions from ZVI due to the polarization effect that weakens Fe-O bonds [17,18]. In parallel, the formation of the complex boosts the electron transfer between the internal Fe(III) and external Fe(II), which weakens the Fe-O bond and accelerates the production of Fe(III) [14,18,19]. Additionally, the formation of soluble complexes between the ligand and Fe prevents precipitation on the surface of ZVI, leaving the electron transfer without a layer of hindrance [20]. In the homogeneous system, the complexation of the iron ion and ligand accelerates the proportion of reactions that generate reactive oxidants, reducing the decomposition of $H_2O_2$ to $H_2O$ [12]. Complexation promotes the conversion of $O_2$ to $O_2^-$, which reacts with protons and electrons to produce surface-bound $H_2O_2$, which also plays an enhancing role in the generation of $H_2O_2$. Chelating agents can form complexes with Fe(III)/Fe(II) at high pH, changing their redox potential of Fe(III)/Fe(II) and maintaining the solubility to prevent precipitation [21]. This provides a ligand site for $H_2O_2$ occupation and promotes the reaction of $H_2O_2$ with Fe(III)/Fe(II) and the generation of oxidation products such as ·OH in Fenton. Finally, the coordination of metals may change the reactive oxidants (e.g., ·OH or Fe(IV)) generated by the interaction of $H_2O_2$ and Fe(II) [22]. In particular, the reactivity of Fe(IV) is predominantly influenced by ligands [23].

Over the years, researchers have reviewed the synthesis and modification of ZVI and its application to the treatment of heavy metal, nitrate, and other contaminants in soil and water [8,12,24,25]. The effect of ZVI characteristics, solution chemistry, and operating conditions on the degradation of contaminants was reviewed by Sun et al. [6]. The application of ZVI and ZVI-based materials for the elimination of heavy metals was reviewed by Zou and his co-workers [26]. Li et al. studied the characterization methods of ZVI for applications in water treatment [27]. Although the modification of ZVI systems using ligands has been mentioned in many research papers, there is no review published focusing on this technology. This review addresses current developments in the degradation of organic pollutants by ligand-enhanced ZVI systems from the perspective of commonly used organic and inorganic chelating agents. The properties of the agents are summarized. The review also clarifies the mechanistic pathways of the technology and proposes its future research directions and application in environmental engineering-related areas.

## 2. Different Chelating Agents

Chelating agents usually consist of more than two functional groups that can be coordinated. Metal atoms replace the hydrogen or metal salt site on the functional group of the chelating agent to form soluble complexes [28,29]. The chelating agents commonly used in the ZVI modification process can be classified into two groups, organic chelating agents and inorganic chelating agents, which are the two perspectives we review in the following sections. In addition, as demonstrated in Table 1, organic pollutants treated by this method include dyes, antibiotics, aromatic chemicals, etc.

### 2.1. Organic Chelating Agents

2.1.1. Aminopolycarboxylic Acids

EDTA

EDTA is a common chelating agent with a wide range of applications in industries, including water softening, metal plating, textile, photography, industrial cleaning, and paper production [30,31]. EDTA is currently the most common aminopolycarboxylic acid

due to its availability and relatively low cost. It contains two amino nitrogen and four carboxyl oxygen groups, making it a tetradentate or hexadentate ligand capable of binding metal ions. EDTA binds to multivalent metal ions in a 1:1 ratio to form strong soluble chelates over a wide pH range [13]. The metal ions are present as anions in the metal-EDTA complex, which induces a shift in the metal ions' redox potential [32]. Moreover, by comparing the stability constants of three aminopolycarboxylic acids, it can be found that the complex of EDTA (log$\beta$ of Fe(II)-complex = 14.3, log$\beta$ of Fe(III)-complex = 25.1) is more stable than that of ethylenediamine-N,N'-disuccinic acid (EDDS) (log$\beta$ of Fe(III)-complex = 20.6) and NTA (log$\beta$ of Fe(II)-complex = 8.05, log$\beta$ of Fe(III)-complex = 15.9) [33-36].

Noradoun et al. [37] discovered that 4-chlorophenol (4-CP) and pentachlorophenol (PCP) were completely degraded in the ZVI/$O_2$/EDTA system at room temperature (Table 1). They speculated that EDTA could effectively complex Fe(II) dissolved from ZVI and enhance $O_2$ activation efficiency. Pan et al. studied the use of EDTA to strengthen the activation of $O_2$ and $H_2O_2$ by ZVI. The ZVI/$O_2$ system removed 10.5% of the sulfamethazine (SMT) in 360 min, while the ZVI/$O_2$/EDTA system reached 70.3% removal of SMT in 60 min [14]. The pre-magnetized ZVI/$H_2O_2$ process removed 19.4% of SMT in 60 min at neutral pH. The presence of 0.1 mM EDTA improved the removal to 90.6% [38]. The authors demonstrated that EDTA inhibited the iron oxide formation on the ZVI surface and promoted the release of iron ions. ZVI was hydroxylated by the ligand and the addition of chelating agents also enhanced Fe(II) recovery from the surface, thus promoting the decomposition of $H_2O_2$ for contaminant removal. Liu et al. used an ultrasonic (US) enhanced ZVI/EDTA/air system to treat 2,4,6-trichlorophenol (2,4,6-TCP). Optimal experimental conditions were obtained as ZVI of 24 g·L$^{-1}$, EDTA of 1.75 mM, initial pH of 7, and 2,4,6-TCP concentration of 50 mg·L$^{-1}$. Under these conditions, the elimination of 2,4,6-TCP reached 98.8% [39]. Zhou and his colleagues studied the degradation of diclofenac (DCF) by a magnetic field (MF)-enhanced ZVI/EDTA system and discussed the effect of varying the initial dose of EDTA from 0.5 to 4.0 mM on the removal of DCF. The rate of dissolved iron accumulation was sped up by increasing the EDTA dose from 0.5 mM to 2.0 mM, but it was not further enhanced with the EDTA dose increasing to 4.0 mM. This is because EDTA can be in competition with the target pollutant for ·OH. In the homogenous Fenton-like system, EDTA served as a competitive contaminant, as evidenced by this [40]. Lu et al. used the combination of an EDTA-enhanced ZVI/Air system and an activated sludge reactor to treat oilfield-produced water [41]. With the support of biotreatment, the removal of hydrolyzed polyacrylamide (HPAM), total petroleum hydrocarbons (TPH), and chemical oxygen demand (COD) exceeded 90%.

Despite many applications in ZVI systems, EDTA is resistant to degradation by biological and chemical processes [42]. Consequently, it is considered as an emerging contaminant with some limitations in its applicability. And given its metal-remobilizing properties, EDTA is gradually losing its edge.

**Table 1.** Degradation Efficiencies of Organic Contaminants by Chelating Agents Enhanced ZVI Systems in Selected Literatures.

| Types | Chelating Agents | Processes | Chemical Doses | pH | Reaction Time | Contaminant Removal |
|---|---|---|---|---|---|---|
| Aminopolycarboxylic acids | EDTA | ZVI/O$_2$ | [ZVI] = 0.3 g· L$^{-1}$, [EDTA] = 2 mM and [SMT] = 5 mg·L$^{-1}$ | 5 | 60 min | 75.3% removal of SMT [43] |
| | | ZVI/O$_2$ | [Granular ZVI] = 50 g· L$^{-1}$, [EDTA] = 0.32 mM and [4-CP] = 1.1 mM | 5.5~6.5 | 4 h | Complete removal of 4-CP [37] |
| | | ZVI/O$_2$ | [Granular ZVI] = 50 g· L$^{-1}$, [EDTA] = 0.32 mM and [PCP] = 0.61 mM | 5.5~6.5 | 70 h | Complete removal of PCP [37] |
| | | pre-ZVI/H$_2$O$_2$ | [ZVI] = 3.6 mM, [EDTA] = 0.1 mM, [H$_2$O$_2$] = 20 μM and [SMT] = 0.018 mM | 7 | 60 min | 90.6% removal of SMT [38] |
| | | US/ZVI/Air | [ZVI] = 24 g· L$^{-1}$, [EDTA] = 1.75 mM, and [2,4,6-TCP] = 50 mg·L$^{-1}$ | 7 | 60 min | 98.8% removal of 2,4,6-TCP [44] |
| | | MF/ZVI | [ZVI] = 0.4 g· L$^{-1}$, [EDTA] = 2 mM and [DCF] = 10 mg·L$^{-1}$ | 5 | 120 min | Almost 72% removal of DCF [40] |
| | NTA | ZVI/O$_2$ | [ZVI] = 0.3 g· L$^{-1}$, [NTA] = 2 mM and [SMT] = 5 mg·L$^{-1}$ | 5 | 60 min | 93.8% removal of SMT [43] |
| | | ZVI/Air | [ZVI] = 15 g, [NTA] = 0.32 mg·L$^{-1}$ and [Caffeic] = 350 mg·L$^{-1}$ | 3.6~3.8 | 180 min | Complete removal of caffeic [44] |
| | | pre-ZVI/H$_2$O$_2$ | [ZVI] = 2 mM, [NTA] = 0.25 mM, [H$_2$O$_2$] = 2 mM and [SMT] = 2 mg·L$^{-1}$ | 7 | 30 min | 99.8% removal of SMT [45] |
| | EDDS | ZVI/O$_2$ | [ZVI] = 0.3 g· L$^{-1}$, [EDDS] = 0.25 mM and [SMT] = 5 mg·L$^{-1}$ | 5 | 60 min | 58.2% removal of SMT [43] |
| | | ZVI/Air | [ZVI] = 20 g· L$^{-1}$, [EDDS] = 0.8 mM, [Air] = 2 L·(min·L) $^{-1}$ and [2,4-DCP] = 100 mg·L$^{-1}$ | 7.37 | 60 min | 99% removal of 2,4-DCP [46] |
| Polyphenols | Gallic acid | nZVI/O$_2$ | [nZVI] = 5 g· L$^{-1}$, [GA] = 10 mg·L$^{-1}$ and [2-CB] = 2 mg·L$^{-1}$ | 5 | 240 min | 65.5% and 59.4% removal of 2-CB at anaerobic and aerobic conditions, respectively [47] |
| | Tea polyphenol extract | mZVI/ H$_2$O$_2$ | [mZVI] = 0.5 g· L$^{-1}$, [Tea polyphenol extract] = 0.1 mM, [H$_2$O$_2$] = 1 mM and [LCM] = 20 mg·L$^{-1}$ | 5.8 | 90 min | 98.85% removal of lincomycin [48] |

| | | | | pH | Time | Result |
|---|---|---|---|---|---|---|
| Polycarboxylates | Citric acid | ZVI/O$_2$ | [ZVI] = 1 g· L$^{-1}$, [CA] = 2 mM and [4-CP] = 0.2 mM | 7 | 60 min | 56.2% removal of 4-CP under dark conditions with lower DO concentrations [49] |
| | | Bi-ZVI/O$_2$ | [ZVI] = 1 g· L$^{-1}$, [CA] = 0.2 mM and [4-CP] = 0.2 mM | 3.6~5.4 | 30 min | 80% removal of 4-CP [50] |
| | | Vis/ZVI/H$_2$O$_2$ | [ZVI] = 12.6 g· L$^{-1}$, [H$_2$O$_2$] = 2.9 mM, [CA] = 1 mM and [RhB] = 21 μM | 7.5 | 60 min | 54% RhB decomposition and 26% COD removal [51] |
| | Oxalic acid | nZVI/O$_2$ | [nZVI] = 1.5 g· L$^{-1}$, [OA] = 100 mM and [PCP] = 5 mg·L$^{-1}$ | 3.18 | 30 min | 83% removal of PCP [52] |
| | | UV/pre-ZVI | [ZVI] = 0.4 g· L$^{-1}$, [OA] = 0.5 mM and [SMT] = 10 mg·L$^{-1}$ | 4 | 60 min | 99.6% removal of SMT [53] |
| | | ZVI/O$_2$ | [ZVI film] = 4 cm$^2$, [OA] = 2.4 mM and [MB] = 9 mg·L$^{-1}$ | 3 | 30 min | 90% removal of MB [54] |
| Polyphosphate | Tetrapolyphosphate | US/ZVI | [ZVI] = 1 g· L$^{-1}$, [TPP] = 0.3 mM and [NOR] = 10 mg·L$^{-1}$ | 7 | 60 min | More than 90% removal of norfloxacin [55] |
| | | nZVI/O$_2$ | [nZVI] = 20 mM, [TPP] = 1 mM and [Atrazine] = 70 μM | 8 | 60 min | Complete removal of atrazine [16] |
| Polyoxometalate | SiW$_{12}$$^{4-}$ | ZVI/ POM | [ZVI] = 0.2 g· L$^{-1}$, [SiW$_{12}$$^{4-}$] = 0.5 mM and [4-CP] = 0.1 mM | 2.5 | 240 min | 40% removal of 4-CP [56] |
| | PW$_{12}$O$_{40}$$^{3-}$ | ZVI/ POM | [ZVI] = 1 mM, [PW$_{12}$O$_{40}$$^{3-}$] = 1 mM and [Phenol] = 10 μM | 2 | 120 min | Almost 62% removal of phenol [57] |
| | | ZVI/ POM | [ZVI] = 1 mM, [PW$_{12}$O$_{40}$$^{3-}$] = 1 mM and [Benzoic acid] = 10 μM | 2 | 120 min | Almost 90% removal of benzoic acid [57] |

Note: US denotes ultrasonic, MF is magnetic field, mZVI represents microscale ZVI, and Vis denotes visible irradiation.

EDDS

As a structural isomer of EDTA, EDDS is a safe and environmentally friendly alternative to EDTA. As it has two chiral C atoms, EDDS possesses three different stereoisomers, i.e., [R,R]-EDDS, [S,S]-EDDS, and [R,S/S,R]-EDDS. [S,S]-EDDS is derived from L-aspartic acid that can be rapidly decomposed by bacteria, rendering it particularly amenable to biodegradation [42,58-60]. Researchers [13,33,58] stated that a favorable pH range for the Fenton process modified by EDDS was 3-9. Fe(III)-EDDS are found in three separate forms in distinct pH ranges: $pH \leq 7$ for Fe(III)-EDDS$^-$ and higher pH levels for Fe(OH)EDDS$^{2-}$ and Fe(OH)$_2$EDDS$^{3-}$. EDDS exhibits photochemical effects and is stable at normal pH levels. By acting as an organic ligand for iron, it produces soluble and active forms of metal and favorably alters its redox potential, enabling the complex to take part in both the oxidation and reduction processes in Fenton systems [58]. EDDS can also increase the absorbance spectrum of iron hydrates in the visible region [61]. Therefore, as a natural biogenic aminopolycarboxylic acid chelator, [S,S]-EDDS is a promising green ZVI modifier [13,62].

Sun et al. [46] investigated the performance of 2,4-dichlorophenol (2,4-DCP) degradation in a ZVI/EDDS/air system. Above 99% removal of 2,4-DCP was accomplished within 1 h in the system under their reaction conditions. In comparison to the EDTA-enhanced ZVI/Air system, the ZVI/EDDS/Air system was more efficient and ecologically friendly in degrading 2,4-DCP at ambient circumstances. Huang et al. [58] examined the role of Fe(III)-EDDS complexes in Fenton-like reactions. After 350 min, it was observed that 99% of BPA had been degraded at 1 mM Fe(III)-EDDS and pH = 6.2. The main factor in determining the process effectiveness was the conversion of Fe(III)-EDDS to Fe(II)-EDDS by $O_2{}^{-}$. Additionally, they found the EDDS-enhanced Fenton reaction was much more effective in removing BPA at pH 8-9 than at pH < 5. Zhang et al. [63] studied the removal of cyclohexanoic acid (CHA) as a model contaminant for naphthenic acids (NAs) in the Fe-EDDS/H$_2$O$_2$ system. A CHA removal rate of 84% was achieved with four continuous additions of 0.74 mM H$_2$O$_2$ and 0.11 mM Fe-EDDS at pH 8. The reaction constant of EDDS with ·OH was $2.48 \pm 0.43 \times 10^9$ M$^{-1}$ s$^{-1}$ (pH = 8), but the rate constant of CHA with ·OH was $4.09 \times 10^9$ M$^{-1}$ s$^{-1}$. Therefore, the scavenging impact of EDDS on ·OH is a major concern that should be taken into account.

This scavenging effect of EDDS on ·OH reduces the effectiveness of treatment processes and limits the application of EDDS. In a follow-up study, Zhang and colleagues discovered that by adding EDDS continuously to the Fenton system, the amount of ·OH that EDDS could scavenge was reduced, which increased the fraction of free radicals for eliminating contaminants. Therefore, continual addition modes are preferred when EDDS is utilized in iron-related reactions [13].

NTA

Nitrilotriacetic acid is one of the simplest aminopolycarboxylic acids. It forms stable and soluble complexes with many metal ions [64,65]. NTA is most frequently used as a detergent builder to chelate calcium and magnesium to stop limescale formation. It can be applied in diverse industries such as pharmaceutical, food, metallurgical, and nuclear decontamination [65-68]. It is a tetradentate-chelating ligand containing four functional groups, a selective N donor and three generic O donors. It forms complexes with metal ions in a 1:1 or 2:1 ratio [42]. NTA is biodegradable, with stronger biodegradability and reducing power than EDDS and EDTA [69]. Metal NTA complexes exhibit rapid photodegradation capabilities [70,71]. The by-products of NTA, such as iminodiacetic acid (IDA) and oxalic acid, impose no adverse impact on the environment [45,72].

Pan and his co-workers investigated the degradation of SMT by NTA-enhanced H$_2$O$_2$ activation by pre-magnetized ZVI (pre-ZVI) at neutral pH. Compared to the H$_2$O$_2$ activation by pre-ZVI, the addition of NTA reduced the degradation time of SMT by half and improved the removal by three times. The corrosion rate ($10^9$) of ZVI was 6.61 cm/s in the

pre-ZVI/NTA/$H_2O_2$ system, which was much higher than that in the ZVI/ $H_2O_2$ (0.73 cm/s), pre-ZVI/$H_2O_2$ (1.47 cm/s), and ZVI/NTA/$H_2O_2$ (2.94 cm/s) systems, indicating that NTA promoted the corrosion of ZVI as the reaction proceeded [45]. Sanchez et al. [44] explored the effect of the ZVI/NTA/Air system on the oxidative degradation of six model organic acids for olive mill wastewater (OMW). Caffeic acid, 4-hydroxyphenylacetic acid, and coumaric acid were all degraded after 180 min, 240 min, and 300 min, respectively. After 360 min, the removal of coumaric acid, tyrosol, and cinnamic acid was 90%, 87%, and 68%, respectively. They also tested the removal of four mixtures of six acids (1000 mg·L$^{-1}$), and a total conversion of 92-99% of the organic compounds was achieved in 360 min. The COD conversion was always above 84%. Wang et al. [43] compared NTA, EDTA and EDDS for ZVI activating oxygen to degrade pollutants. They found that NTA reacted with $O_2$ at a faster rate and argued that the caging effect of EDTA and EDDS hindered the reaction rate between Fe(II) and $O_2$. Luca and co-workers [73] proposed that iron complexes of EDTA and NTA were more stable than those of oxalic acid and tartaric acid. At neutral pH, NTA was superior with regard to biodegradability, micropollutant removal, and total organic carbon (TOC) contribution compared to EDTA. Moreover, the NTA and ·OH rate constant at pH 8 was reported to be $4.77 \pm 0.24 \times 10^8$ M$^{-1}$ s$^{-1}$, which reflects the lower reactivity between the two [74].

Overall, NTA is safer than EDTA and EDDS and appears to be a better alternative for the effective enhancement of ZVI, Fenton, or Fenton-like systems for degrading wastewater contaminants at near-neutral pH conditions. However, its use is controversial, with researchers reporting that it shows moderate toxicity to humans and mammals [75]. Moreover, NTA with fewer ligand sites may require a larger dose to chelate iron compared to EDDS [13].

### 2.1.2. Polyphenols

Polyphenols are benzene derivatives that contain one or more hydroxyl groups coupled with additional substituents connected to one or more aromatic rings [21,76]. Plants, fruits and vegetables, nuts, soy beans, tea, and cocoa are all rich in polyphenols. When deprotonated, o-dihydroxy groups like catechol and gallate are primarily responsible for the metal chelating properties of polyphenols. As metal ions with a preference for octahedral geometries, Fe(III) and Fe(II) can cooperate with three catechol or gallate groups with a large stability constant [77]. However, polyphenolic compounds have a wide range of structural characteristics, the majority of them have numerous sites for binding to Fe(III), and their complexes with iron can have various coordination properties and are pH-dependent. This makes it particularly challenging to analyze the properties of polyphenolic chemicals' interaction with Fe(III) [78]. Polyphenols are believed to promote the reduction of Fe(III) to Fe(II) and enhance the production of oxidizing substances such as ·OH [79,80]. Many studies have now reported the positive effect of polyphenols, such as catechol and gallol, on the efficiency of Fenton reactions.

Wang and his co-workers [47] studied the role of tannic acid and gallic acid on the enhancement of 2-chlorobiphenyl (2-CB) degradation by nZVI at aerobic circumstances. They found that the two polyphenols worked as electron shuttle mediators, facilitating the electron transfer from the nZVI surface to $O_2$ and increasing the production of Fe(II) and $H_2O_2$, thus improving the generation of ·OH, a key substance leading to the removal of 2-CB. Ouyang et al. [48] compared the degradation effect of six promoters, i.e., tea polyphenols (TP), ascorbic acid (AA), hydroxylamine hydrochloride (HA), EDTA, oxalic acid, and sodium tripolyphosphate on lincomycin (LCM) in the mZVI/$H_2O_2$ system. Under the optimal reaction conditions, the LCM degradation by the mZVI/$H_2O_2$ system with TP increased by 73% compared with the mZVI/$H_2O_2$ system alone. The presence of TP facilitated LCM removal through the conversion of Fe(III) to Fe(II) and ion chelation without apparently altering the pH. They argued that TP has strong antioxidant properties, strong chelating properties and weak acidity, has a high unit removal rate (URR), and is more universal than other chelating agents.

Researchers have concluded that polyphenols have the following effects on ·OH produced in Fenton/-like reactions: (i) forming complexes with iron ions, and modifying the potential of iron-polyphenols converting $H_2O_2$ to ·OH, (ii) boosting the production of ·OH by strengthening the reversion of Fe(III) to Fe(II), and (iii) scavenging ·OH [21,81]. Actually, polyphenols' reduction property is commonly exploited in synthesizing different metal nanoparticles (NPs). Polyphenol-rich plant extracts are increasingly being used for green production of iron nanoparticles [82-84]. Despite various studies that have shown that polyphenols can promote the effectiveness of ZVI and Fenton systems, certain polyphenolic compounds have the reverse inhibitory impact on these systems under particular conditions. For example, iron chelation of polyphenolic compounds like tannic acid inhibits the Fenton reaction [85,86]. To summarize, many aspects of the use of polyphenolic chemicals in iron environmental chemistry are yet unknown and need to be investigated further.

### 2.1.3. Polycarboxylates

Citric Acid

As a naturally occurring $\alpha$-hydroxy tricarboxylic acid, citric acid (CA) is widely distributed in nature and acts as a Fe(III) transporter in biological systems [87]. A compound with a 1:1 molar ratio between iron and citrate was described by Lanford and Quinan [88]. Other researchers have confirmed the presence of three species: $[Fe(Cit)]^0$, $[Fe(HCit)]^+$, and $[Fe(Cit)(OH)]^-$ [89,90]. However, there are studies that have reported the presence of 2:2 complexes [91], mononuclear dicitrates [92], and dinuclear complexes [93]. It is reported that citrate ions could function as ligands to form stable complexes with Fe(III), thereby strengthening the Fenton reaction to generate reactive oxygen species (ROS) at appropriate pH [50,94].

Xu et al. [49] investigated the synergistic oxidative removal of 4-CP in a CA-enhanced nZVI system. Under dark conditions with lower dissolved oxygen concentrations, the ZVI/CA system degraded 56.2% of 4-CP within 60 min. They suggested Fe(II)[Cit]⁻ that was adsorbed on ZVI surface may react with $H_2O_2$ more efficiently than Fe(II), accelerating the electron transport from ZVI to $O_2$ and the generation of ·OH. Tso et al. [95] reported that CA has a great complexing ability and that the passive layer of ZVI can form soluble complexes through a ligand-facilitated solubilization process. The concentration of the citrate ligand boosts the pace of the reaction, but large quantities of oxalic acid induce precipitation of the complex to the ZVI surface and impede its activity. Gong and his co-workers [50] utilized citric acid to promote the aerobic removal of 4-CP by bismuth-modified ZVI nanoparticles (Bi-ZVI). The Bi-ZVI/CA/$O_2$ system removed 80% of 4-CP within 30 min with high dichlorination efficiency. They identified that CA combined with Fe(III) to generate a ligand complex (i.e., Fe(III)Cit), which improved the production of large amounts of ROS in the Bi/ZVI system at both aerobic and anaerobic conditions. Under visible irradiation for 1 h, Hong et al. [51] observed 54% removal of Rhodamine B (RhB) and 26% removal of COD in the ZVI/$H_2O_2$/CA system at a fairly low level of dissolved iron (5.4 μmol·L⁻¹). In the citrate-free system, RhB removal reached 31% in 2.5 h. Qian and his colleagues [96] investigated the coupled application of up-flow anaerobic sludge blanket (UASB), sequencing batch reactor (SBR), and the homogeneous Fenton-like system promoted by citric acid. The addition of citric acid significantly improved the degradation of antibiotics in swine wastewater, particularly the removal of trimethoprim (TMP), which is not biodegradable by biotreatment, by up to 80%. From this study, it is evident that citric acid has great potential for practical applications in areas related to ZVI and Fenton.

The reaction rate constants between CA and ·OH were reported as $1.2 \times 10^8$ M⁻¹ s⁻¹ (pH 3), $1.5 \times 10^8$ M⁻¹ s⁻¹ (pH 3.6), $2.4 \times 10^8$ M⁻¹ s⁻¹ (pH 6), and $3.2 \times 10^8$ M⁻¹ s⁻¹ (pH 6.6) [97]. They explained that deprotonation of CA to a more reactive dissociate is responsible for the increase in reactivity with pH. Fe-citrate has a low stability constant, and therefore more agent is needed to chelate all the iron in the treatment systems. When the CA content

is high, the citrate ion reacts with ·OH to form 3-hydroxyglutaric acid radicals and 3-oxoglutaric acid radicals, and this inhibits the oxidation of the substrate by ·OH [51]. The addition of CA lowers the final pH of the system owing to the relatively low $pK_a$ ($pK_1$ = 3.00, $pK_2$ = 4.60, and $pK_3$ = 5.80 [98]), which may affect the reactivity of ZVI [13,95]. Therefore, it is important to consider the impacts of variables like dosage and pH when applying CA to ZVI systems.

Oxalic Acid

Oxalic acid (OA) is a low molecular weight dicarboxylic acid with significant acidity, chelating ability, and reducing power [99]. For the decomposition of different organic contaminants, oxalate is widely utilized to generate photosensitive complexes such as ferrioxalate with ferric ions. The ferrioxalate complex ($[Fe(C_2O_4)_3]^{3-}$) is a combination of three bidentate oxalate ions with an iron center. Photolysis of the complexes produces Fe(II) and oxalate radicals. The quantum yield was 1.24 at 300 nm (pH = 2 and 6 mM of ferrioxalate) [100,101]. Further, studies have shown that oxalate radicals react with $O_2$ to form $HO_2·$, $O_2^-$, and their conjugate acid. $O_2^-$/ $HO_2·$ can disproportionate or react with Fe(II) to form $H_2O_2$ [102].

Pan et al. [53] explored SMT degradation by the UV/pre-ZVI/Oxalate system and obtained 99.6% removal of SMT at OA of 0.5 mM, pre-ZVI of 0.4 $g·L^{-1}$, SMT of 10 $mg·L^{-1}$, and pH = 4. OA serves as an inhibitor of iron (hydrogenated) oxides formation, enhancing the hydroxylation of the ZVI surface. Pan et al. [14] also investigated the reusability of ZVI and oxalate, reporting that 6.3 $mg·L^{-1}$ of SMT had 58.2% elimination in the ZVI/Oxalate system after three cycles of treatment. The rate constant k ($10^3$) for five consecutive runs was 13.6, 12.5, 11.6, 10.7, and 9.5 $min^{-1}$, respectively, demonstrating the potential of the ZVI/Ox system for application in practical wastewater treatment. Tso and Shih [95] investigated the impact of oxalate on the surface properties of ZVI. Along with the removal of the passive layer from the ZVI surface by a complicated process at 15 mM OA, they also stated that the oxalate-Fe complex itself might have the ability to degrade trichloroethylene (TCE). Ou et al. [52] observed that the nZVI/OA system achieved 83% degradation of pentachlorophenol at an initial pH of 3.18. Oxalate-Fe complexes on nZVI particles reduced the production of ferrous (hydro) oxides that precipitated on the surface of nZVI. Moreover, as OA has strong complexation and pH buffering properties, it allowed for the dechlorination of PCP by nZVI with an efficiency of 89%. Wei and his co-workers [54] prepared ZVI films via electrodeposition and found that the ZVI/OA/DO system decolored methylene blue (MB) more effectively than the EDTA system and was stable for long-term use. MB decolorization rates in the system are strongly influenced by factors such as pH and OA concentration. In a certain range, with the lower the pH and the higher the OA concentration, the better the decolorization effect. Doumic et al. [103] found that in the case of biotreatment combined with a photo-Fenton system, the presence of iron oxalate complexes inhibited iron precipitation and led to substantial mineralization and complete decolorization of the simulated textile wastewater.

Compared to other organic chelating agents like CA, EDTA, and EDDS, OA and its dissociated species react relatively slowly with ·OH, with reaction rates ranging from 1.4 × $10^6$ $M^{-1}$ $s^{-1}$ [104] to 1 × $10^7$ $M^{-1}$ $s^{-1}$ [97], which is lower than that of CA by 1 or 2 orders of magnitude. This indicates that the reagent has a low ·OH scavenging rate and little competition with the target contaminant. However, similar to the CA, the low stability constant of oxalate-Fe complexes suggests that high doses of OA are required to chelate all the iron. The presence of OA may also lead to pH decline in the reaction system, as indicated by the low $pK_a$ of OA ($pK_1$ = 1.23 and $pK_2$ = 4.19 [105]).

*2.2. Inorganic Chelating Agents*

As we previously mentioned, the presence of organic ligands (EDTA, citrate, etc.) can increase the oxidant yield in the ZVI system. However, organic ligands are expensive and can have unfavorable effects, such as reactive oxygen species depletion and decreased reaction efficiency due to decomposition of the ligands. Therefore, scientists have noticed that inorganic ligands with high stability and low cost may be more advantageous in boosting reactive oxidants generation in ZVI systems [16].

2.2.1. Polyphosphates

Polyphosphates, an inorganic food additive and a traditional source of energy, are linear polymers consisting of tens to hundreds of orthophosphate residues linked by phosphonic anhydride bonds [106,107]. They can be generated through organism-mediated processes and are very common. Polyphosphates are relatively safe, low cost, and environmentally friendly [16,108,109]. The main role of polyphosphate in Fenton and ZVI systems is reported to prevent iron precipitation at neutral pH by forming soluble complexes through iron chelation and lowering the potential of Fe(III)/Fe(II) [110]. It also improves ZVI electron availability by facilitating electron transport from the ZVI core to surface-bound/soluble Fe(III)-TPP, which in turn regenerates Fe(III)-TPP to Fe(II)-TPP [16].

Zhang et al. [110] selected sodium tripolyphosphate (STPP) as a promoter to enhance Fe(II) activation of $O_2$. They found that >80% of p-nitrophenol (PNP) was eliminated in 40 min in the Fe(II)/$O_2$/STPP system. $O_2^-$, the main oxidant for direct decomposition of PNP, reduced PNP to p-aminophenol. This allowed the STPP-enhanced system to remove organic pollutants that are difficult to react directly with ·OH. Collaborative degradation of norfloxacin (NOR) in the ultrasound-enhanced ZVI/TPP system (US/ZVI/TPP) was investigated by Zhou et al. [55]. Compared with three popular organic chelating agents (EDTA, EDDS, and DTPA), the TPP-mediated system was more effective in degrading NOR with relatively low doses of ZVI and ligands in a pH operating range of 3-9. They obtained optimized initial conditions as NOR of 10 mg·L$^{-1}$, TPP of 0.3 mM, ZVI of 1 g·L$^{-1}$, and initial pH = 7. Wang et al. [16] studied the impact of TPP on the activation of molecular oxygen to degrade atrazine by Fe@Fe$_2$O$_3$ nanowires, a specific nZVI synthesized by their group without magnetic stirring. At pH = 8, 1 mmol·L$^{-1}$ TPP addition enabled the atrazine decomposition to reach about 100% within 60 min. In the TPP-enhanced Fe@Fe$_2$O$_3$ system, Fe(II) concentration increased from 400 µmol·L$^{-1}$ to 800 µmol·L$^{-1}$ and Fe(III) concentration decreased from 240 µmol·L$^{-1}$ to 136 µmol·L$^{-1}$ within 60 min. This Fe(III) reduction process proved that TPP addition was able to inhibit hydrogen precipitation through proton confinement, allowing more electrons to remain from the iron core. Kim and his co-workers [111] found that the nZVI/TPP/$O_2$ system presented oxidant yields similar as that of the Fe(II)/TPP/$O_2$ system in terms of iron consumption, suggesting that nZVI acted primarily as a Fe(II) source. Methanol was used to scavenge reactive oxidants and its oxidation product, HCHO, was quantified as one of the indicators of oxidant yield. The oxidant yield (D[HCHO]/D[Fe(II)]) was close to 70% in the Fe(II)/$O_2$/TPP system. As this value exceeded the theoretical maximum yield of HCHO (D[HCHO]/D[Fe(II)] = 50%), they argued that a series of single electron transfer for Fe(II) oxidation by oxygen was not sufficient to explain it, and there may be a more direct pathway for producing reactive oxidants (e.g., Fe(II) oxidation by oxygen to Fe(IV)).

Researches have indicated that polyphosphate concentrations can be sustained in the system, which implies that polyphosphate can be collected appropriately, reducing the possible adverse effect of the ligands on the environment [112]. Additionally, TPP is already widely used as a food additive. For these reasons, TPP can be considered as an ideal ligand for ZVI environmental remediation applications.

### 2.2.2. Polyoxometalate

As a metal-oxygen cluster anion, polyoxometalate (POM) can be reduced reversibly without undergoing structural change [113]. POM is photo-excited by the absorption of UV photons, and the photo-excited POM can be reduced by extracting an electron from the organic substrate. The reduced POM can be re-oxidized by dissolved oxygen and it can act as an electron shuttle between the electron donor and acceptor to resist oxidation and catalyze the redox reaction [56,114,115]. In addition, POM can mediate the transport of electrons from ZVI to $O_2$, ultimately generating ·OH through in situ $H_2O_2$ production [56].

Lee et al. [56] reported the application of two POMs, $SiW_{12}O_{40}^{4-}$ and $PW_{12}O_{40}^{3-}$, in ZVI systems. Organic pollutants such as bisphenol A, N-nitrosodimethylamine, phenolic compounds, sulfamethoxazole, and terephthalic acid were effectively degraded in the $ZVI/O_2/SiW_{12}O_{40}^{4-}$ system. These two POM ions act to accelerate the electron transfer between the two phases and promote $H_2O_2$ generation, which in turn initiates ·OH radicals via Fenton-type reactions, triggering a series of oxidation reactions. Mylon et al. [116] confirmed that the oxidation capacity of nZVI could be enhanced by increasing the dose of sodium polyoxotungstate ($Na_3PW_{12}O_{40}$) at low pH. They concluded that the mechanism for the enhanced oxidation capacity of nZVI consisted of two aspects: (i) POM competed with $H_2O_2$ for electrons in ZVI, thus increasing the concentration of $H_2O_2$; and (ii) reduced POM promoted the circulation from Fe(III) to Fe(II), thus enhancing reactions in the homogeneous system. Lee et al. [57] proposed enhancement mechanisms of nZVI or dissolved Fe(II) systems by POM at neutral pH. The complexes of iron and POM prevented precipitation on the surface of ZVI or in the native solution. Moreover, the yields of oxidants of the $POM/ZVI/O_2$ and $POM/Fe(II)/O_2$ systems were close to the theoretical maximum at pH 7. Zhang's group [117] synthesized an insoluble Fe(III)-containing POM ($Fe^{III}AspPW_{12}$), which was used to degrade 4-CP in a Fenton-like system. TOC was completely removed within 120 and 60 min from 100 mg·$L^{-1}$ 4-CP solution at neutral pH under dark sedimentation and irradiation, respectively, with the corresponding $H_2O_2$ utilization of 77% and 83%. This study verified the possibility of applying POM for modifying iron-based materials in environmental applications.

POMs as inorganic ligands are relatively non-toxic and resistant to oxidation, offering POM systems an edge over organic ligand systems regarding efficiency and environmental security. POM is expensive and its reuse is necessary for cost-effective applications in a range of treatment systems [57]. However, POMs hydrolyze at neutral and alkaline pH, and therefore, the stability of the material needs to be improved, limiting their application [23,118]. Therefore, the potential application of POM in ZVI and Fenton systems deserves in-depth study.

### 3. Conclusions and Perspectives

This review describes the current application of ligand-enhanced ZVI systems in organic pollutant removal from the perspective of organic and inorganic chelating agents. Generally, organic chelating agents like EDTA have excellent metal complexing properties and performance in enhancing the efficiency of ZVI systems, but their toxicity and secondary contamination limit their application. Inorganic chelating agents, like POM, are generally non-toxic and do not react with the free radicals generated, so they have a significant advantage over organic ligands. The role of inorganic chelating agents on ZVI systems deserves further studies.

The efficiency of the ligand-enhanced ZVI systems will need to be evaluated in future studies by using a broader spectrum of contaminants. In addition, the properties, generation, and transformation of reactive oxidants generated in the ligand-enhanced ZVI systems are not fully understood in current studies and require further exploration. Toxicological and cost-effective studies of chelating agents are also needed. In practical applications, various methods (such as designing different reactors and combining with other

processes) that enhance ZVI systems with different chelating agents need to be further investigated.

**Author Contributions:** Conceptualization, Writing—original draft preparation, Writing—review and editing, Q.C.; Conceptualization, Supervision, Funding acquisition, Resources, Writing—review and editing, M.Z.; Resources, Y.P.; Conceptualization, Supervision, Funding acquisition, Resources, Writing—review and editing, Y.Z. All authors have read and agreed to the published version of the manuscript.

**Funding:** This research was funded by Key Project of Natural Science Foundation of Tianjin (No. 21JCZDJC00320), the National Natural Science Foundation of China (Nos. 21773129, 22176102 and 21976096), and the Fundamental Research Funds for the Central Universities.

**Conflicts of Interest:** The authors declare no conflict of interest.

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
