# Peer review of "Ligand-Enhanced Zero-Valent Iron for Organic Contaminants Degradation: A Mini Review"

_processes, doi:10.3390/pr11020620_

Round 1

Reviewer 1 Report

The review article is well written and presents the state of the art on the use of zero-valent iron (ZVI) in AOPs. I only suggest highlighting the scientific contribution of this review in the introduction of the manuscript and whether other works have already been done in this direction, highlighting the progress achieved with this compilation.

Author Response

Reviewer #1: The review article is well written and presents the state of the art on the use of zero-valent iron (ZVI) in AOPs. I only suggest highlighting the scientific contribution of this review in the introduction of the manuscript and whether other works have already been done in this direction, highlighting the progress achieved with this compilation.

Response: We thank the Reviewer for the comment and suggestion. We have revised the sentences in the Introduction to highlight the scientific contribution of this review: “Although the modification of ZVI systems using ligands has been mentioned in many research papers, there is no review published focusing on this technology. This review addresses current developments in the degradation of organic pollutants by ligand-enhanced ZVI systems from the perspective of commonly used organic and inorganic chelating agents. The properties of the agents are summarized. The review also further clarifies the mechanistic pathways of the technology and proposes its future research directions and application in environmental engineering related areas.”

Reviewer 2 Report

The review is of considerable interest and well done. I recommend it to be published after a minor revision.

1. The Authors should also proofread their manuscript (some spelling and grammar errors).

2. All equation should be revised, which exist as an image and contain some minor error.

3.Some publications are suggested to refer to improve the quality of the manuscript, such as: https://doi.org/10.1016/j.colsurfa.2021.127753, https://doi.org/10.1016/j.surfin.2022.102006,  https://doi.org/10.1016/j.heliyon.2022.e09652.

4. The conclusion is too long and also not targeted to the important aspects described in the review; please rephrase it.

Author Response

The review is of considerable interest and well done. I recommend it to be published after a minor revision.

1. The Authors should also proofread their manuscript (some spelling and grammar errors).

Response: Thanks for the comment and suggestion. We have checked the manuscript and corrected spelling and grammar errors in it.

2. All equation should be revised, which exist as an image and contain some minor error.

Response: We thank the Reviewer for the suggestion. Equations in the manuscript have been checked and revised.

3. Some publications are suggested to refer to improve the quality of the manuscript, such as:

https://doi.org/10.1016/j.colsurfa.2021.127753,

https://doi.org/10.1016/j.surfin.2022.102006,

https://doi.org/10.1016/j.heliyon.2022.e09652.

Response: Thank you for the recommendation. We have read these articles on different composite materials carefully, but they are less relevant to this review and are not cited. We will actively refer to them in our future work.

4. The conclusion is too long and also not targeted to the important aspects described in the review; please rephrase it.

Response: Thanks for the suggestion. In the conclusion part, we summarize the advantages and disadvantages of the application of organic and inorganic chelating agents in zero-valent iron systems and give an outlook on the technology. Following your suggestion, we have revised some sentences in the conclusion.

Reviewer 3 Report

The review article entitled " Ligand -enhanced zero-valent iron for organic contaminants degradation " has a good contribution to the field. It well written and all sections are clearly described. However I have some comments:

1- Zero-valent iron(ZVI) environmental applications should be illustrated as a diagram and be added in the section of the introduction.

2- The ZVI technologies for organic contaminants degradation like ZVI combined with physical m chemical and biological methods should be added as a new title after the inorganic chelating agent.

3- The author must consider letter writing as a subscript or superscript like H2O2  mgL-1  M-1 s etc...

Finally the review is aaccepted for publication after minor revision.
